# Protective efficacy of six recombinant proteins as vaccine candidates against *Echinococcus granulosus* in dogs

Guoqing Shao[1☯], Ruiqi Hua[1☯], Hongyu Song[1], Yanxin Chen[1], Xiaowei Zhu[1], Wei Hou[2], Shengqiong Li[2], Aiguo Yang[2]*, Guangyou Yang[1]*

1 Department of Parasitology, College of Veterinary Medicine, Sichuan Agricultural University, Chengdu, Sichuan Province, P. R. China, 2 Sichuan Center for Animal Disease Prevention and Control, Chengdu, Sichuan Province, P. R. China

☯ These authors contributed equally to this work.
* aiguoyang163@163.com (AY); guangyou1963@126.com (GY)

**Data Availability Statement:** All relevant data are within the manuscript and its Supporting Information files.

## Abstract

### Background

Cystic echinococcosis (CE) is caused by the infection of *Echinococcus granulosus* sensu lato (*E. granulosus* s.l.), one of the most harmful zoonotic helminths worldwide. Infected dogs are the major source of CE transmission. While praziquantel-based deworming is a main measure employed to control dog infections, its efficacy is at times compromised by the persistent high rate of dog re-infection and the copious discharge of *E. granulosus* eggs into the environment. Therefore, the dog vaccine is a welcome development, as it offers a substantial reduction in the biomass of *E. granulosus*. This study aimed to use previous insights into *E. granulosus* functional genes to further assess the protective efficacy of six recombinant proteins in dogs using a two-time injection vaccination strategy.

### Methods

We expressed and combined recombinant *E. granulosus* triosephosphate isomerase (r*Eg*TIM) with annexin B3 (r*Eg*ANXB3), adenylate kinase 1 (r*Eg*ADK1) with *Echinococcus* protoscolex calcium binding protein 1 (r*Eg*EPC1), and fatty acid-binding protein (r*Eg*FABP) with paramyosin (r*Eg*A31). Beagle dogs received two subcutaneous vaccinations mixed with Quil-A adjuvant, and subsequently orally challenged with protoscoleces two weeks after booster vaccination. All dogs were sacrificed for counting and measuring *E. granulosus* tapeworms at 28 days post-infection, and the level of serum IgG was detected by ELISA.

### Results

Dogs vaccinated with r*Eg*TIM&r*Eg*ANXB3, r*Eg*ADK1&r*Eg*EPC1, and r*Eg*FABP-*Eg*A31 protein groups exhibited significant protectiveness, with a worm reduction rate of 71%, 57%, and 67%, respectively, compared to the control group (*P* < 0.05). Additionally, the vaccinated groups exhibited an inhibition of worm growth, as evidenced by a reduction in body

**Funding:** This work was funded by the Key Technology R&D Program of Sichuan Province, China (Grant No. 2022YFN0013 to G.Y). The funders had no role in study design, data collection and analysis, decision to publish, or preparation of the manuscript.

**Competing interests:** The authors have declared that no competing interests exist.

length and width ($P < 0.05$). Furthermore, the level of IgG in the vaccinated dogs was significantly higher than that of the control dogs ($P < 0.05$).

## Conclusion

These verified candidates may be promising vaccines for the prevention of *E. granulosus* infection in dogs following two injections. The r*Eg*TIM&r*Eg*ANXB3 co-administrated vaccine underscored the potential for the highest protective efficacy and superior protection stability for controlling *E. granulosus* infections in dogs.

## Author summary

Cystic echinococcosis (CE), a neglected parasitic zoonosis of global health significance, poses a persistent threat to humans and imposes substantial economic burdens. Dogs are the main source for human CE infections, although the implementation of praziquantel-based deworming programs targeting dogs, certain regions continue to experience high human CE incidence, highlighting the need for alternative preventive strategies such as dog vaccines. Various recombinant vaccines have exhibited promising efficacy; however, no commercially licensed vaccines are currently available for field use due to two key challenges. Firstly, the existing dog vaccine immunization regimen necessitates three injections to achieve adequate protection. Secondly, the vaccine's protective efficacy displays considerable variability among the dogs in vaccinated groups. To address these limitations, we employed the *Escherichia coli* expression system to produce six potential vaccine candidates. Additionally, we introduced a novel immune protocol involving a two-protein combined vaccine administered via a two-time injection in dogs. Employing this immunization protocol, the r*Eg*TIM&r*Eg*ANXB3 co-administrated vaccine demonstrated exceptional stability, exhibiting minimal standard deviation and resulting in a notable 71% reduction in worm burden. Implementation of this two-protein combined vaccine with only two injections not only mitigates the CE infection risk to human and livestock but also represents the more cost-effective and practical solution ever for dog owners.

## Introduction

Cystic echinococcosis (CE) is a zoonotic parasitic disease caused by the ingestion of *Echinococcus granulosus* sensu lato (*E. granulosus* s.l.) eggs [1]. CE poses a significant public health concern given its substantial morbidity in livestock and humans worldwide, most notably in remote areas that lack adequate hygiene and infrastructure [2,3]. Recent estimates indicate that the worldwide prevalence of CE constitutes a considerable burden, with an annual new incidence of approximately 188,000 cases, thereby contributing to 1,097,000 disability-adjusted life years [4]. In the life circle of *E. granulosus*, dogs play a crucial role as the major definitive hosts, serving as the main source of human CE infection [5]. Thus, reducing *E. granulosus* infections in dogs holds great potential as a method for CE control. Currently, the implementation of praziquantel-based deworming has been a cornerstone measure to fight against *E. granulosus* infection in dogs [6,7]. However, the deworming is often impractical because of the requirement frequency dosing (at least occur every 3 months) which is often not achieved in endemic areas [8,9]. In Rio Negro and Morocco, prior studies have demonstrated that vaccinating the intermediate host with EG95 yielded an immune protection rate of 97%, whereas

praziquantel treatment in owned dogs showed limited efficacy in reducing sheep incidence [10,11]. Remarkably, analysis reveals that a dog deworming program lasting for many years in China, despite significant investment of human resources and financial means, has resulted in an escalation of overall infections of human CE [12]. Thus, vaccination of dogs assumes a better substitution of deworming, as it can substantially reduce the parasite biomass within the dog intestine [13]. Moreover, mathematical models indicates that a 75% reduction in worm burden through dog vaccination could potentially eliminate the transmission of *E. granulosus* to human and livestock [14]. Additionally, considering that there is a much smaller population of dogs in comparison to intermediate hosts, dog vaccine would provide a more practical and feasible measure [6,15].

Initially, crude proteins were prepared from protoscoleces (PSCs) and cysts for the purpose of canine vaccination. However, the vaccination of dogs with such crude proteins has resulted in limited immune protection against *E. granulosus* infection [16]. Recently, the development of *E. granulosus* vaccines has shifted towards utilizing recombinant subunit antigens expressed by bacterial systems which have now become the most used approaches [17]. Among the vaccine candidates that have been thoroughly investigated, the recombinant proteins *Eg*A31-*Eg*Trp, *Eg*M, and *Eg*HCDH have exhibited promising potential in reducing worm burden and egg production [18–20]. However, in the current phase of dog vaccine development, two obstacles impede the availability of licensed vaccines for field use. Firstly, the worm burdens in dogs of the vaccinated groups show massive variation. Secondly, in comparison to the two-time vaccine injection (including two doses for initial immunization and a booster per year) used for newborn lambs with EG95 (a licensed vaccine for sheep) [21], the current dog recombinant vaccines require three injections to achieve limited and unstable protection. This discrepancy in the number of required injections translates into a much higher financial cost. Given these challenges, it is crucial to enhance the stability of protection and optimize the immune protocol for dog vaccine against *E. granulosus* infection.

Our previous investigations revealed that the use of acquired recombinant proteins, including r*Eg*TIM, r*Eg*ANXB3, r*Eg*ADK1, and r*Eg*EPC1 [22–25], alongside the fused protein r*Eg*FABP-*Eg*A31, represent promising vaccine candidates. As previous studies, the use of two antigen components can enhance the protection stability of parasite vaccines [26]. To further advance our research, we have introduced an innovative approach that involves the development of a two-protein combined vaccine (which applied two different systems, one with the co-administration of two proteins separately, and one with the administration of two fused proteins) administered via a two-time injection immunization protocol in dogs. In this study, we present our findings on the efficacy and immune response elicited by these vaccine candidates.

## Materials and methods

### Ethics statement

The animal study was reviewed and approved by the Animal Care and Use Committee of Sichuan Agricultural University (SYXK2019-187). All animal procedures used in this study were carried out in accordance with the Guide for the Care and Use of Laboratory Animals (National Research Council, Bethesda, MD, USA) and recommendations of the ARRIVE guidelines (https://www.nc3rs.org.uk/arrive-guidelines). All methods were carried out in accordance with relevant guidelines and regulations.

### Animals and parasites

A total of 24 beagles, aged between four and five months, were obtained from the Beagles Breeding Center, Sichuan Institute of Musk Deer Breeding (Sichuan, China). The beagles

consisted of an equal number of males and females, and all the dogs were dewormed with albendazole and praziquantel half month prior to the primary vaccination. Subsequently, the dogs were randomly assigned to their respective experimental groups. All the dogs were housed in Specific Pathogen Free conditions with a controlled quarantine facility and fed a standardized diet of commercial dog food with access to tap water.

PSCs were collected from the fresh livers with the hydatid cysts of sheep slaughtered in abattoir of Sichuan province, China. PSC viability and molecular genotyping was assessed using methods as described in our previous studies [27]. PSCs with a viability higher than 95% and identified as *E. granulosus* sensu stricto (s.s.; G1-G3) (which were responsible for over 97% of CE infections in China [3]) were considered for the challenge process.

## Protein expression and purification

Recombinant *Eg*TIM, *Eg*ANXB3, *Eg*ADK1, and *Eg*EPC1 were obtained as previously described [22–25]. We expressed and purified *Eg*FABP fused with a partial gene of *Eg*A31 as follows. Specific primers (sense primer 5′-CGC GGA TCC ATG GAG CCA TTC ATC GGT-3′ and antisense primer 5′-CCG GAA TTC CAT CCC TCT TGA GTA GGT TCT-3′) were designed for PCR based on the *Eg*FABP gene sequence (GenBank accession number: XM_024494196.1), which introduced *BamH* I and *EcoR* I restriction enzyme sites (underlined). Specific primers (CCG GAA TTC AAT GCT GAG GTT CTT CGT-3′ and antisense primer 5′- CCC AAG CTT CTA CAT GAT ACT GGT TGC ACG-3′) were also designed based on the partial sequence of the EgA31 gene (GenBank accession number: AF067807), which introduced the *EcoR* I and *Hind* III restriction enzyme sites.

The PCR product was purified prior to digestion with restriction enzymes, ligated to the expression vector, pET-32a (Novagen, Madison, USA), and transformed into *E. coli* BL21 (DE3) (Cowin Biotech, Beijing, China). A positive *E. coli* clone with the correct DNA sequence was cultured in LB medium, and the expression of fused r*Eg*FABP-*Eg*A31 was induced using 1 mM IPTG at 16°C for 10 h. The r*Eg*FABP-*Eg*A31 protein with a poly-histidine tag was affinity purified from bacterial lysates using the Ni-NTA His-tag resin (Qiagen, Hilden, Germany). The purified r*Eg*FABP-*Eg*A31 was analyzed by 12% SDS-PAGE, and the concentration of recombinant proteins was determined using a BCA protein assay kit (Beyotime, Jiangsu, China).

## Vaccination and parasite challenge

The randomized allocation of the 24 Beagle dogs into four groups was conducted as described in Table 1 according to the requirements of World Association for the Advancement of Veterinary Parasitology [28]. For each vaccination, 100 μg of each soluble recombinant protein and 1 mg Quil-A (Superfos Biosector, Denmark) in 1 mL PBS were mixed and left to stir overnight at 4°C prior to administration. The dogs were administered a primary and booster vaccination via subcutaneous injection, with a 2-week interval between each vaccination. Two weeks after the booster vaccination, all dogs were orally challenged with 70,000 PSCs. Four weeks post-

**Table 1. Group of dogs used in this study.**

| Group | Dogs | Vaccine (per dog) |
|---|---|---|
| Control | 6 | 1 mg Quil-A + 1 mL PBS |
| r*Eg*TIM&r*Eg*ANXB3 | 6 | 100 μg r*Eg*TIM + 100 μg r*Eg*ANXB3 + 1 mg Quil-A + 1 mL PBS |
| r*Eg*ADK1&r*Eg*EPC1 | 6 | 100 μg r*Eg*ADK1 + 100 μg r*Eg*EPC1 + 1 mg Quil-A + 1 mL PBS |
| r*Eg*FABP-*Eg*A31 | 6 | 200 μg r*Eg*FABP1-*Eg*A31 + 1 mg Quil-A + 1 mL PBS |

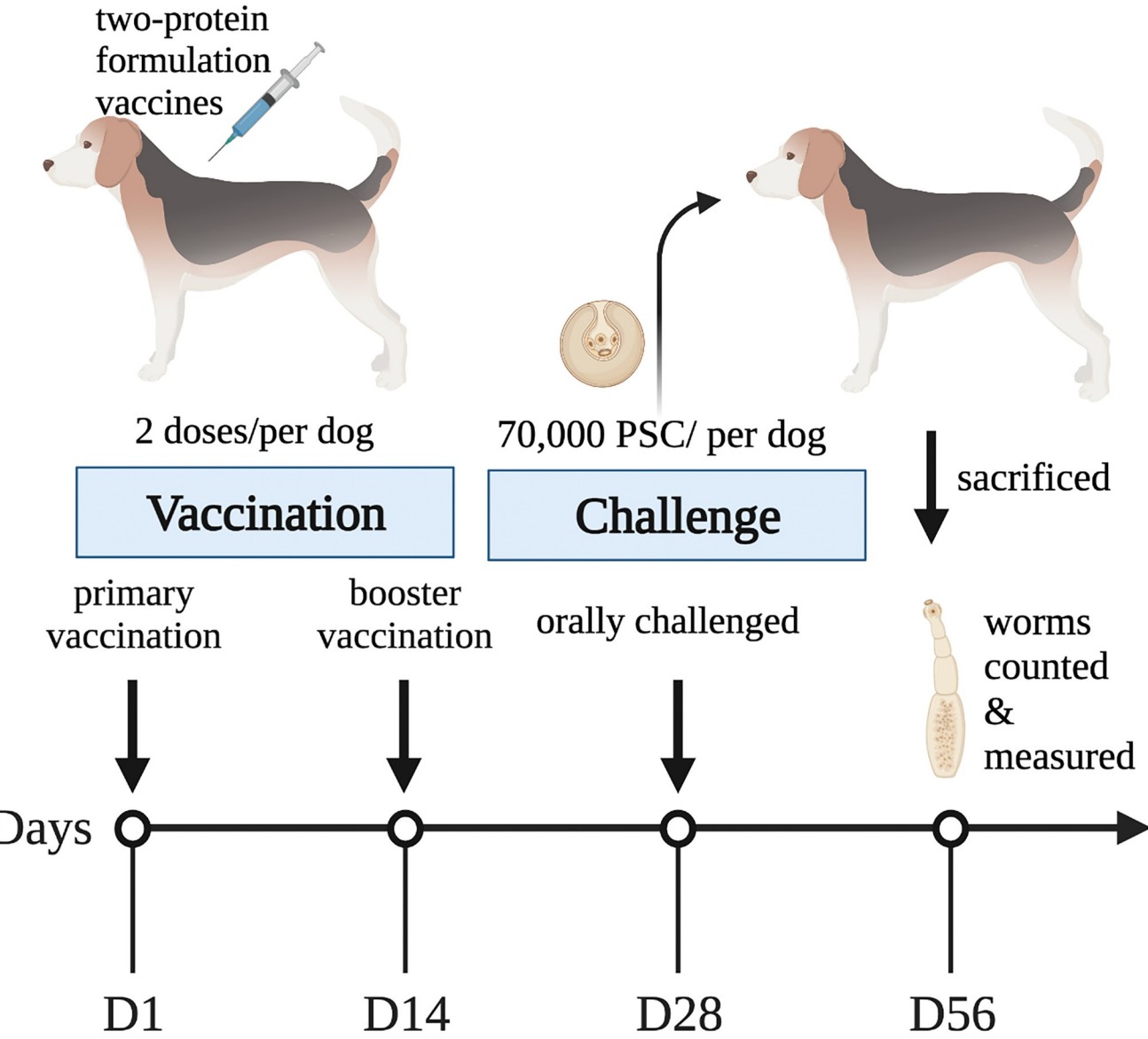

**Fig 1.** Schematic representation (Created with BioRender.com) of the vaccination process and parasite challenge in the present study.

infection, all dogs were humanely euthanized and underwent necropsy (Fig 1). Tapeworms present in the small intestine of each dog were collected and recorded. The body length and width were also measured in 300 randomly selected tapeworms per group. The weekly sera extractions were performed for further IgG detection.

### ELISA detection of serum IgG

ELISA conditions were optimized using a checkerboard titration method with the vaccine antigens and sera. The *Eg*TIM, *Eg*ANXB3, *Eg*ADK1, *Eg*EPC1, and *Eg*FABP-*Eg*A31 purified proteins were diluted in PBS to 5 μg/mL. The ELISA plates were coated with diluted antigen solutions and incubated overnight at 4˚C. The plates were subsequently washed three times with PBST and blocked with 5% skim milk for 2 h at 37˚C. After washing the wells three times,

100 μL of the serum samples (diluted 1:100 in PBST) were added and incubated at 37˚C for 1 h. The HRP-labeled rabbit anti-dog IgG (diluted 1:3000, Solarbio, Beijing, China) was added and incubated for 1.5 h at 37˚C. The wells were subsequently washed again and incubated with TMB (Tiangen, Beijing, China) substrate at 37˚C for 20 min. Finally, the reaction was stopped with 100 μL of 2 M $H_2SO_4$, and the optical density was measured at 450 nm.

### Data collection and analysis

We employed a Mann-Whitney U test to statistically compare the worm burden in experimental groups with the control group. Data were analyzed with SPSS software v 22.0, and the average values were used to calculate the reduction in worm numbers. A *P*-value less than 0.05 was considered statistically significant.

## Results

### Expression and purification of recombinant proteins

The recombinant protein preparations, including r*Eg*ANXB3, r*Eg*TIM, r*Eg*ADK1, r*Eg*EPC1, and fused r*Eg*FABP-*Eg*A31 were successfully expressed and purified. Our results demonstrated that the expressed proteins were of the expected molecular weights, with r*Eg*ANXB3 at approximately 53 kDa, r*Eg*TIM at 46 kDa, r*Eg*ADK1 at 40 kDa, r*Eg*EPC1 at 26 kDa, and the fused r*Eg*FABP-*Eg*A31 at 60 kDa, as confirmed by SDS-PAGE analysis. (Fig 2).

### Vaccination of dogs with recombinant proteins

In comparison with the control group, the vaccination groups induced significant protection of the reduction in worm burdens and inhibition of worm growth, as detailed in Table 2. In terms of worm reduction, our results showed that the r*Eg*TIM&r*Eg*ANXB3, r*Eg*ADK1&r*Eg*EPC1, and r*Eg*FABP-r*Eg*A31 vaccine groups exhibited superior protective efficacy, with a remarkable 71% (*P* = 0.004), 57% (*P* = 0.006), and 67% (*P* = 0.004) reduction in the worm burden after two subcutaneous injections, compared to the control group. As for the inhibition of worm growth, our results revealed that the worm growth of vaccine groups was significantly suppressed in 24% to 32% (*P* < 0.05) of the length and width of the worm bodies (Fig 3 and S1 Table).

### Dog serum IgG titers detection by ELISA

Our findings revealed that the experimental groups exhibited a remarkable increase in IgG titers compared to the control group (Fig 4). Notably, the highest levels of IgG were observed during the first week following the second booster vaccination, indicating a robust and prompt immune response. Furthermore, the IgG levels maintained a steady plateau within a certain range over time for weeks. However, the duration of the immune response had not yet been determined.

## Discussion

In this study, we employed a two-protein combined formulation, which was administered to dogs using a two-time injection immunization protocol to evaluate the protectiveness of six vaccine candidates against *E. granulosus* s.s. infections. To the best of our knowledge, this is the first trial conducted in dogs that has evaluated this immunization strategy in the context of dog vaccine against *E. granulosus* s.s. tapeworms.

Based on our previous studies and those of others, six proteins (r*Eg*TIM, r*Eg*ANXB3, r*Eg*ADK1, r*Eg*EPC1, r*Eg*FABP and r*Eg*A31) that have been demonstrated to play a crucial role

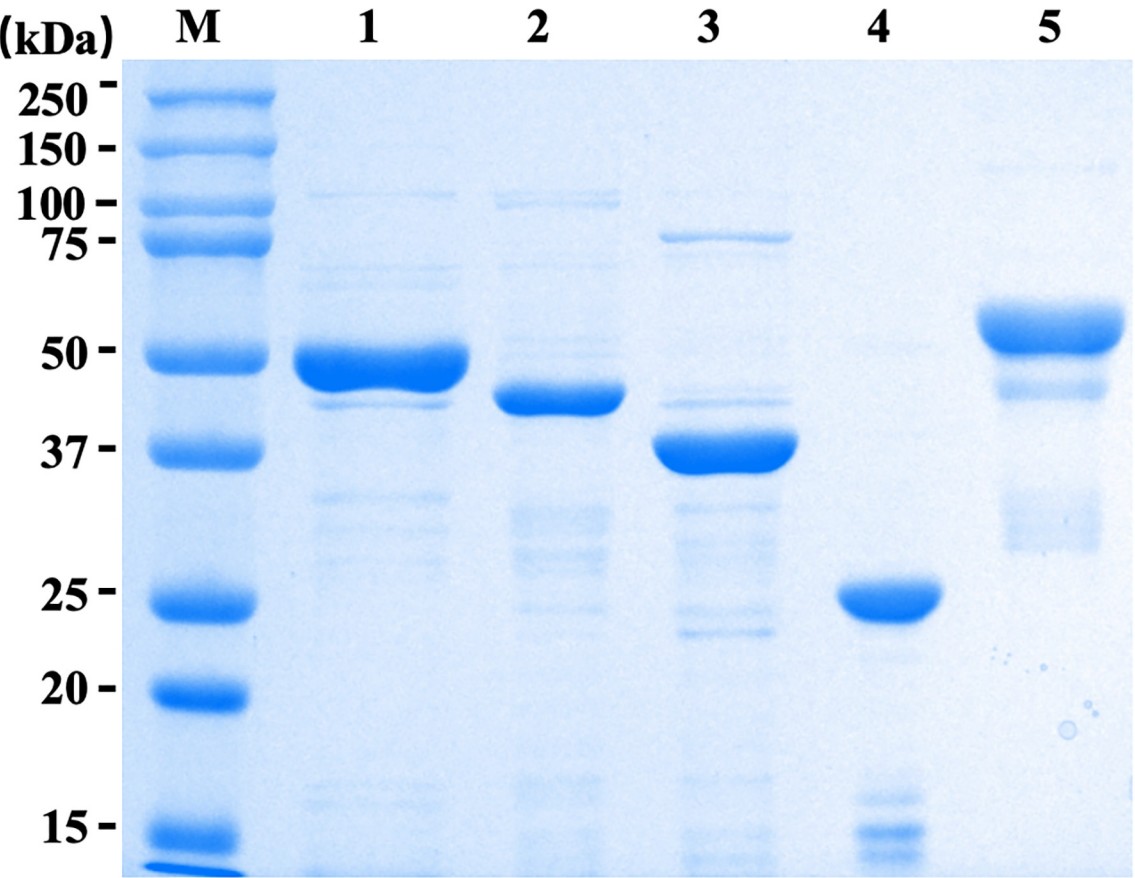

**Fig 2. SDS-PAGE analysis of the purified recombinant proteins used in this study.** M = molecular mass marker in kDa; lane 1 = purified r*Eg*ANXB3; lane 2 = purified r*Eg*TIM; lane 3 = purified r*Eg*ADK1; lane 4 = purified r*Eg*EPC1; and lane 5 = purified r*Eg*FABP-*Eg*A31.

in the development and survival of *E. granulosus* were selected as potential vaccine candidates [22–25]. *Eg*TIM is a pivotal enzyme in glycolysis, serving as the primary energy source for *E. granulosus*, and also known to be involved in several immune regulation functions [22,29]. In addition, we found that annexins of *E. granulosus* s.s. might play an important role in parasite-host interactions and demonstrated that the level of *Eg*ANXB3 transcription was increased from PSCs to adults [23,30]. This indicates that it has a potential role during the adult stages of infection in dogs. *Eg*ADK1 is a crucial enzyme involved in the regulation of adenosine triphosphate metabolism in *E. granulosus*. *Eg*ADK1 has been suggested as a potential vaccine candidate due to its role in cellular energy balance [24,31,32]. Furthermore, *Eg*EPC1 was selected as a pattern protein due to its robust immunogenicity and ability to interact with the host through secretion [33]. *E. granulosus* *Eg*A31 and *Eg*FABP were previously identified as potential vaccine candidates [34,35]. Specifically, recombinant *Eg*A31 has been shown to enhance the cell-mediated immune response in dogs, indicating its potential as a promising vaccine antigen [34]. Tapeworms are incapable of synthesizing fatty acids, and instead rely on acquiring fats from the host through the fatty acid binding protein [35]. Based on these collective results, we hypothesized that the inclusion of r*Eg*TIM&r*Eg*ANXB3, r*Eg*ADK1&r*Eg*EPC1, and r*Eg*FABP-*Eg*A31, which had not been previously evaluated, might enhance the immune response and facilitate deworming efforts in the vaccinated group.

**Table 2.** *E. granulosus* worm burdens on day 28 post-infection in dogs vaccinated with vaccines mixed with Quil-A.

| Group | Dog no. | No. of worms | Average | SD | Reduction (%) [a] | P value |
|---|---|---|---|---|---|---|
| Control | D1 | 18885 | 17423 | 1466 | - | - |
| | D2 | 16585 | | | | |
| | D3 | 15305 | | | | |
| | D4 | 18265 | | | | |
| | D5 | 18868 | | | | |
| | D6 | 16630 | | | | |
| A (r*Eg*TIM&r*Eg*ANXB3) | A1 | 9232 | 5110 | 2933 | 71% | 0.004 |
| | A2 | 685 | | | | |
| | A3 | 6545 | | | | |
| | A4 | 4515 | | | | |
| | A5 | 6235 | | | | |
| | A6 | 3445 | | | | |
| B (r*Eg*ADK1&r*Eg*EPC1) | B1 | 2665 | 7428 | 6037 | 57% | 0.006 |
| | B2 | 2985 | | | | |
| | B3 | 16400 | | | | |
| | B4 | 2735 | | | | |
| | B5 | 13435 | | | | |
| | B6 | 6345 | | | | |
| C (r*Eg*FABP-*Eg*A31) | C1 | 935 | 5749 | 5320 | 67% | 0.004 |
| | C2 | 11553 | | | | |
| | C3 | 5735 | | | | |
| | C4 | 908 | | | | |
| | C5 | 2497 | | | | |
| | C6 | 12867 | | | | |

[a] Reduction (%) = (average worm number in control group—average worm number in the experiment group) / average worm number in control group × 100%.

Adjuvants play a critical role in vaccine development by enhancing antigen-specific immune responses [36]. Out of several adjuvants (i.e., Quil-A, Freund's adjuvant, and ISCOMs) commonly employed in canine vaccines [17], Quil-A has demonstrated similar protective efficacy to that of antigens mixed with Freund's adjuvant or ISCOMs [37]. By contrast, Quil-A is comparatively more cost-effective, stable, and safe compared to that of other adjuvants. Thus, in the present study, we opted to employ Quil-A emulsified with vaccine antigens to immunize dogs, and no adverse effects were observed among the subjects. Our findings suggested that the Quil-A adjuvant might represent a safe option for canine vaccines.

The serum IgG response was likely associated with the host protection against infection [37]. Our data revealed that these recombinant proteins were successful in stimulating and sustaining the immune response in dogs over a period of weeks following primary vaccination. Meanwhile, our findings demonstrated that each of the three vaccine antigen groups significantly enhanced the protective immunity of dogs against *E. granulosus* infection, as evidenced by a reduction in worm burden and growth.

Compared to the current dog vaccine candidates, including r*Eg*Ms (with worm reduction rate 33%-99%), r*Eg*HCDH (87%, not significant for $P = 0.400$) (administered via three subcutaneous injections) [19,37], and the *Salmonella* oral vaccine *Eg*A31-*Eg*Trp (70%-80%) [20], the vaccine candidates evaluated in our study demonstrated a favorable protective efficacy after two subcutaneous injections. Our findings revealed several advantages in terms of vaccine candidates: (a) the vaccination protocol with a reduction of one injection compared to existing

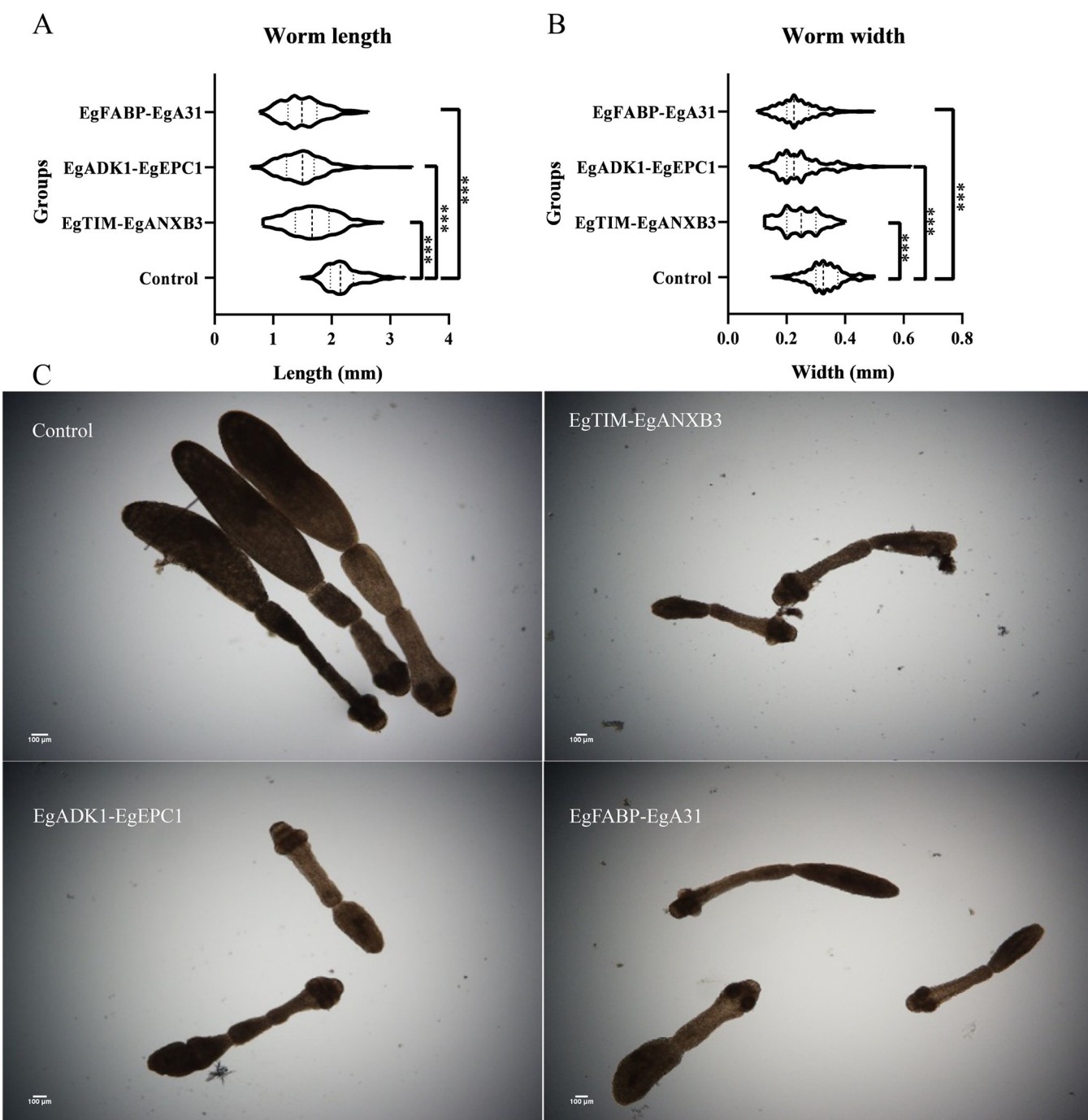

**Fig 3. *E. granulosus* worm burdens and sizes on day 28 post-infection in dogs.** (A and B) The violin plot for the length and width of 300 worms in each group (***$P < 0.001$). (C) The size of each group of tapeworms under the microscope (×100). Representative images of the tapeworms in the control group, r*Eg*TIM&r*Eg*ANXB3 group, r*Eg*ADK1&r*Eg*EPC1 group, and r*Eg*FABP-*Eg*A31 group.

canine vaccine candidates; (b) a smaller standard deviation compared to single antigen vaccines (e.g., r*Eg*M4, r*Eg*M9, and r*Eg*HCDH); (c) compared to the *Salmonella* vaccine EgA31-EgTrp, the vaccine candidates evaluated in our study demonstrated better data stability and reduced the biological risk by suppressing the worm development. However, some

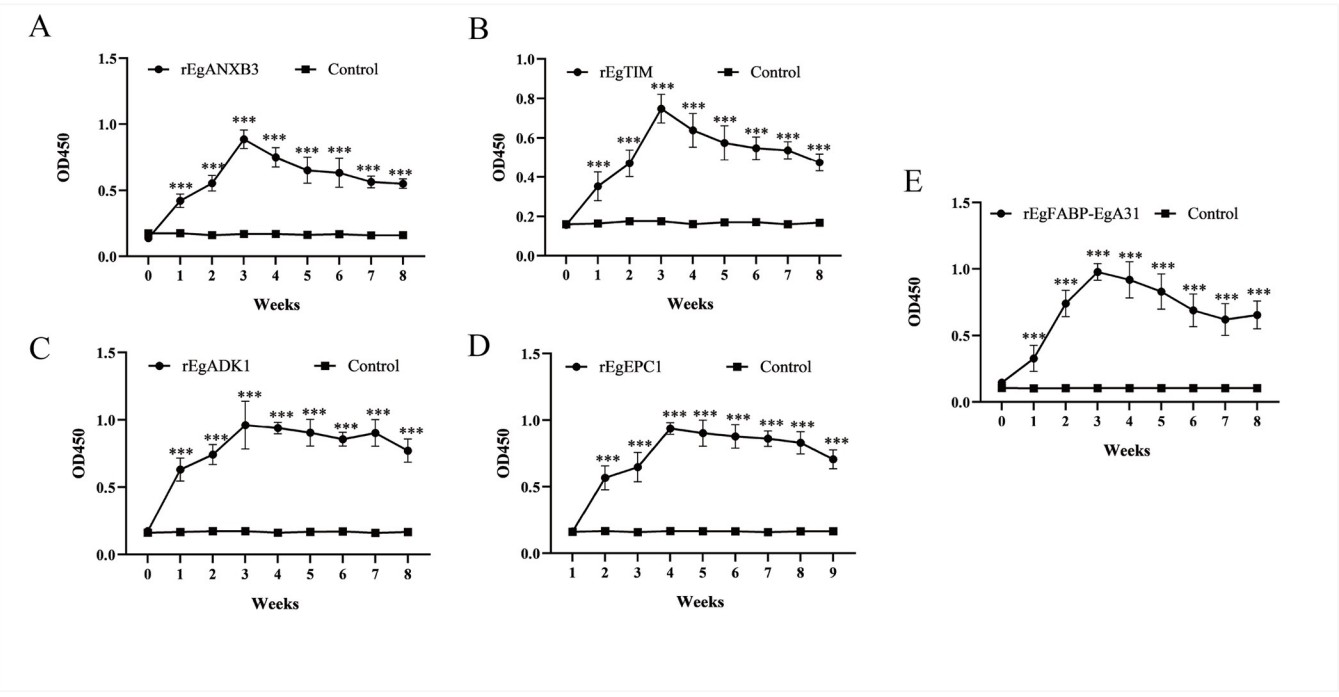

**Fig 4. Serum levels of IgG generated in response to recombinant proteins as detected by ELISA.** The serum was diluted 1 in 100. The serum levels of IgG against recombinant r*Eg*ANXB3 (A), r*Eg*TIM (B), r*Eg*ADK1 (C), r*Eg*EPC1 (D), and recombinant fusion protein, r*Eg*FABP-*Eg*A31 (E), as detected by ELISA. *P* value less than 0.05 was significant between the control and experiment groups (***$P < 0.001$).

limitations were observed. The worm burdens in dogs of the vaccinated groups showed massive variation. However, compared to single antigen vaccines, such as r*Eg*M and r*Eg*HCDH, the data showed a smaller standard deviation, that suggested the two antigen vaccines might demonstrate better protection stability against parasite infection. Overall, our innovative strategy of combining two recombination proteins via two subcutaneous injections in dogs presented a highly practical and cost-effective solution for the development of canine vaccines against *E. granulosus* tapeworms.

## Conclusions

Our findings suggested that this immunization protocol (two-protein combined vaccine administered via a two-time injection) could induce effective immune protection against *E. granulosus* infections in dogs. The r*Eg*TIM&r*Eg*ANXB3 provided 71% protection against *E. granulosus* infection in dogs and exhibited superior protection stability, with a minimal standard deviation. Therefore, our research provided potential candidate vaccines and cost-effective immune protocol against *E. granulosus* infection.

## Supporting information

**S1 Table. The detailed data for length and width of *E. granulosus* s.s. worms at day 28 post-infection in dogs of vaccination and control groups.** Reduction (%) = (average worm size in control group—average worm size in the experiment group) / average worm size in control group × 100.
(XLSX)

## Acknowledgments

We sincerely thank all the participants who participated in the dog vaccine development. We also thank the workers of Beagles Breeding Center for their assistance with dog management.

## Author Contributions

**Conceptualization:** Aiguo Yang, Guangyou Yang.

**Data curation:** Guoqing Shao, Hongyu Song.

**Formal analysis:** Guoqing Shao, Ruiqi Hua.

**Funding acquisition:** Aiguo Yang, Guangyou Yang.

**Investigation:** Guoqing Shao, Ruiqi Hua, Hongyu Song, Yanxin Chen, Xiaowei Zhu, Wei Hou, Shengqiong Li.

**Software:** Guoqing Shao, Ruiqi Hua, Hongyu Song.

**Supervision:** Aiguo Yang, Guangyou Yang.

**Writing – original draft:** Guoqing Shao, Ruiqi Hua, Hongyu Song.

**Writing – review & editing:** Guoqing Shao, Ruiqi Hua.

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
