## [Decision Letter · Decision Letter 0]

29 Aug 2023

Dear Dr. Yang,

Thank you very much for submitting your manuscript "Protective efficacy of six recombinant proteins as vaccine candidates against Echinococcus granulosus in dogs" for consideration at PLOS Neglected Tropical Diseases. As with all papers reviewed by the journal, your manuscript was reviewed by members of the editorial board and by several independent reviewers. The reviewers appreciated the attention to an important topic. Based on the reviews, we are likely to accept this manuscript for publication, providing that you modify the manuscript according to the review recommendations. 

All the reviewers commented that the study was interesting and the manuscript well written. Minor edits and consideration of the reviewer's points are required before the manuscript is suitable for publication.

Sincerely,

Krystyna Cwiklinski, PhD

Academic Editor

Eva Clark

Section Editor

All the reviewers commented that the study was interesting and the manuscript well written. Minor edits and consideration of the reviewer's points are required before the manuscript is suitable for publication.

Reviewer's Responses to Questions

**Key Review Criteria Required for Acceptance?**

**Methods**

-Are the objectives of the study clearly articulated with a clear testable hypothesis stated?

-Is the study design appropriate to address the stated objectives?

-Is the population clearly described and appropriate for the hypothesis being tested?

-Is the sample size sufficient to ensure adequate power to address the hypothesis being tested?

-Were correct statistical analysis used to support conclusions?

-Are there concerns about ethical or regulatory requirements being met?

Reviewer #1: It is a paper that can mark an important change in CE control

We understand that the introduction can be improved by ordering the different possible interventions. Include more background. For example, there are programs where the PZQ has been successful in decreasing the number of human cases. The problems pointed out by the authors are correct. Secondly, vaccination in sheep also presents a history of success in stopping transmission. Also, the problems mentioned regarding the size of the population to be vaccinated are correct. So it would be better to present vaccination in dogs as a new alternative that in principle can complement the previous ones. I would avoid using the term eradication. It is better to talk about eliminating the transmission to man.

The experiences of use of EG95 include 2 doses and a booster per year. or more. See papers of Rio Negro or Morocco 

in results, can the authors indicate the exact values of P ?

There should be a mention that the duration of the immune response has not yet been determined

Reviewer #2: The submitted article named “Protective efficacy of six recombinant proteins as vaccine candidates against Echinococcus granulosus in dogs” is a well written article that explores the use of three vaccine formulations (each composed of two recombinant antigens) to limit the establishment of Echinococcus granulosus sensu lato infection in dogs. As dogs plays an important role in the E. granulosus life cycle as the definitive host, a successful vaccine could indirectly protect both farm animals and humans, as both act as intermediate hosts. Moreover, as there are a lot fewer dogs than farm animals in rural populations, dogs vaccination programs are usually consider as a more cost- and time-effective strategy.

This article is easy to read, well written and self-explanatory. The methods are consistent with the objectives, and the obtained results support the conclusion. 

There are some key comments I would like to point out 

1) E. granulosus s.l. genomic differences:

E. granulosus is a complex of parasites divided into nine genotypes grouped in four species. There are significant genomic differences between them, and not all genotypes are fully sequenced. It would be advice to genotype the protoscoleces used for infections / obtained worms (or at least do a bibliographical research of the genotypes present on sheeps in China) and evaluate weather the recombinant antigens have any amino acid sequence difference between the canonical gen (G1) and the genotypes present. Not knowing this could be playing against you, as the dogs’ infections might be with G1 and G5, for example, and the vaccine only protect against G1. 

2) Measurement of mucosal immunity

The consensus when developing a vaccine against an intestine parasite is that a significant mucosal immunity should be achieved. In fact, the salmonella vaccine using A31-Trp has shown that this response is greatly protective. It is known that oral vaccines can generate better mucosal immunity. Why was the selection of the immunization scheme subcutaneous, and why was not a measurement of serum/mucosal IgAs.

3)Other considerations

When describing the vaccine protocol in dogs, it should be mentioned that weekly sera extractions were performed for further IgG quantification.

Reviewer #3: (No Response)

**Results**

-Does the analysis presented match the analysis plan?

-Are the results clearly and completely presented?

-Are the figures (Tables, Images) of sufficient quality for clarity?

Reviewer #1: (No Response)

Reviewer #2: The results are well presented and clearly understood. The figures present well the obtained results. Figure 4 seems to be a bit blurry, but it might be because of the PDF. The overall article is straightforward and the obtained results are derived from the hypothesis and conducts to the conclusions. The only weak point of the research that I can point out is the lack of sera and mucosal IgA quantification, to propose a link between mucosal immune response and the protection generated

Reviewer #3: (No Response)

**Conclusions**

-Are the conclusions supported by the data presented?

-Are the limitations of analysis clearly described?

-Do the authors discuss how these data can be helpful to advance our understanding of the topic under study?

-Is public health relevance addressed?

Reviewer #1: (No Response)

Reviewer #2: The overall article is straightforward, as it is a evaluation of the protection generated by tree vaccine formulations. There are no speculative conclusion to be derived from the results, as its shown how much the worms burden is reduced after vaccination. The article is original, it can have a real application into the E. granulosus vaccine development

Reviewer #3: (No Response)

**Editorial and Data Presentation Modifications?**

Reviewer #1: (No Response)

Reviewer #2: I would recommend this article to be accepted with Minor Revisions, mostly text-wise.

Reviewer #3: (No Response)

**Summary and General Comments**

Reviewer #1: (No Response)

Reviewer #2: I would like to recommend some text modifications to improve the overall reading experience:

First of all, when the recombinant antigens are first introduced, I would explicitly say the whole name of the protein (eg. EgANXB3: Annexin B3) as it might help to do parallelism with other pathogen vaccines. 

Secondly, when mentioning the current dog vaccine (line 111), the composition of the available vaccine/vaccines should be mentioned, as its not clear if it is a crude antigen vaccine, recombinant vaccine or purified vaccine. This also makes the “cost” referred in line 113 not easily associated to the vaccine development, as a crude antigen vaccine industrial processes in unthinkable due to E. granulosus not being a cultivable parasite. Also, there should be more context around EG95.

On the Salmonella A31-Trp vaccine, they already use a combination of two antigens, making your overall innovating two-protein strategy, although interesting, not fully innovating

Then describing the “two-protein combined vaccine administration” it should reflect that there are two different system applied, one with the co-administration of two proteins separately, and one with the administration of two fused proteins.

It would be advice to compare the worm reduction values here obtained with the previously reported

On line 266, Salmonella A31-Trp vaccine did not have any associated risk, the genetically modified salmonella cannot regain their pathogenesis mechanisms, and the biodistribution of salmonella was previously analyzed, and shown to be safe. The biological risk that is mentioned is the possibility of developed worms (in either control or vaccinated dogs) to produce oncospheres that may infect the handlers, which is also present in this article

On line 239 and 277, the “rEgTIM-rEgANXB3” should not include the “-“ as both antigens are not conjugated, but co-administrated

Quil-A® is a registered trademark and should include the ®

Reviewer #3: Congratulations to the authors for this interesting work, especially as this is a recombinant vaccine trial based on a combination of two proteins at the same time, knowing that dog vaccination is a very important element in the control of this zoonosis.

The comments made do not undermine the quality of this work.

1- Line 130: The dogs were treated with Albendazol and Prziquantel, with no residual effect. Have you taken fecal samples to ensure that the dogs are free of E. granulosus parasites or eggs? 

2- Line 139: On what basis did you select the proteins used in the vaccine? Why these sequences and not others? 

3- Line 169: After the intestinal scraping, especially in the control dogs, did you find any eggs in the adult parasite worms collected?

4- Line 171: in table 1, why did you use a 200µg dose of the EgFABP1-EgA31 vaccine and not a 100µg dose like the others? 

5- Line 220: On what basis did you combine the different proteins?

PLOS authors have the option to publish the peer review history of their article (what does this mean?). If published, this will include your full peer review and any attached files.

Reviewer #1: Yes: EDMUNDO LARRIEU

Reviewer #2: Yes: Sebastian Miles

Reviewer #3: Yes: Fatimaezzahra Amarir

Figure Files:

Data Requirements:

Reproducibility:

References

---

## [Decision Letter · Decision Letter 1]

9 Oct 2023

Dear Dr. Yang,

We are pleased to inform you that your manuscript 'Protective efficacy of six recombinant proteins as vaccine candidates against Echinococcus granulosus in dogs' has been provisionally accepted for publication in PLOS Neglected Tropical Diseases.

Best regards,

Krystyna Cwiklinski, PhD

Academic Editor

Eva Clark

Section Editor

The authors have addressed the comments raised by the reviewers. The manuscript is now suitable for publication in PLoS NTD.

Reviewer's Responses to Questions

**Key Review Criteria Required for Acceptance?**

**Methods**

-Are the objectives of the study clearly articulated with a clear testable hypothesis stated?

-Is the study design appropriate to address the stated objectives?

-Is the population clearly described and appropriate for the hypothesis being tested?

-Is the sample size sufficient to ensure adequate power to address the hypothesis being tested?

-Were correct statistical analysis used to support conclusions?

-Are there concerns about ethical or regulatory requirements being met?

Reviewer #1: yes

Reviewer #2: (No Response)

**Results**

-Does the analysis presented match the analysis plan?

-Are the results clearly and completely presented?

-Are the figures (Tables, Images) of sufficient quality for clarity?

Reviewer #1: yes

Reviewer #2: (No Response)

**Conclusions**

-Are the conclusions supported by the data presented?

-Are the limitations of analysis clearly described?

-Do the authors discuss how these data can be helpful to advance our understanding of the topic under study?

-Is public health relevance addressed?

Reviewer #1: yes

Reviewer #2: (No Response)

**Editorial and Data Presentation Modifications?**

Reviewer #1: (No Response)

Reviewer #2: (No Response)

**Summary and General Comments**

Reviewer #1: good paper

Reviewer #2: The present article is ready to be accepted. The authors comply with all the suggestions me and the others reviewers ask for, which made the manuscript easier to read and understand. The current version of the manuscript have no mayor or minor revisions needed, it has a clear importance to overall science, and can have a big impact on one health

PLOS authors have the option to publish the peer review history of their article (what does this mean?). If published, this will include your full peer review and any attached files.

Reviewer #1: **Yes: **yes

Reviewer #2: **Yes: **Sebastian Miles

---

## [Editor Report · Acceptance letter]

18 Oct 2023

Dear Dr. Yang,

We are delighted to inform you that your manuscript, "Protective efficacy of six recombinant proteins as vaccine candidates against Echinococcus granulosus in dogs," has been formally accepted for publication in PLOS Neglected Tropical Diseases.

Best regards,

Shaden Kamhawi

co-Editor-in-Chief

Paul Brindley

co-Editor-in-Chief
